# The Pharmacological Treatment of Pediatric Vertigo

**DOI:** 10.3390/children9050584

**Published:** 2022-04-20

**Authors:** Pasquale Viola, Gianmarco Marcianò, Alessandro Casarella, Davide Pisani, Alessia Astorina, Alfonso Scarpa, Elena Siccardi, Emanuele Basile, Giovambattista De Sarro, Luca Gallelli, Giuseppe Chiarella

**Affiliations:** 1Unit of Audiology, Regional Centre of Cochlear Implants and ENT Diseases, Department of Experimental and Clinical Medicine, Magna Graecia University, 88100 Catanzaro, Italy; pasqualeviola@unicz.it (P.V.); davidepisani@gmail.com (D.P.); alessiaastorina7@gmail.com (A.A.); 2Clinical Pharmacology and Pharmacovigilance Unit, Department of Health Science, School of Medicine, University of Catanzaro, Mater Domini Hospital, 88100 Catanzaro, Italy; gianmarco.marciano@libero.it (G.M.); al.cas1993@gmail.com (A.C.); emanuele.basile1082@virgilio.it (E.B.); desarro@unicz.it (G.D.S.); gallelli@unicz.it (L.G.); 3Department of Medicine and Surgery, University of Salerno, 84081 Baronissi, Italy; alfonsoscarpa@yahoo.it; 4ENT Unit, Maria Vittoria Hospital, 10144 Torino, Italy; elenasiccardi@yahoo.it; 5Research Center FAS@UMG, Department of Health Science, Magna Graecia University, 88100 Catanzaro, Italy; 6Medifarmagen SRL, Department of Health Science, Magna Graecia University, 88100 Catanzaro, Italy

**Keywords:** children, vertigo, drugs, pediatric, dizziness, vestibular, balance

## Abstract

Vertigo in children is a challenging topic. The lack of dedicated trials, guidelines and papers causes inhomogeneity in the treatment of vertigo in children. Meniere’s disease, migraine equivalents, vestibular neuritis, paroxysmal positional benign vertigo (BPPV), persistent postural-perceptual dizziness (PPPD) and motion sickness may affect children with various degrees of incidence and clinical severity compared to adults. Several drugs are proposed for the management of these conditions, even if their use is subordinated to the child’s age. In this review, we summarize the existing evidence related to the use of drugs for this clinical condition in children as a start point for new trials, stating the urgent need for international guidelines.

## 1. Introduction

Vertigo and dizziness are terms used to describe a wide range of symptoms related to disorders of motion perception and coordination of the body. Vertigo is the sensation of self-motion when no self-motion is occurring or the sensation of distorted self-motion during an otherwise normal head movement. Dizziness is the sensation of disturbed or impaired spatial orientation without a false or distorted sense of motion. Postural symptoms are balance symptoms related to the maintenance of postural stability, occurring only while upright (seated, standing, or walking) [1,2]. In comparison with the adult population, in children, vertigo and dizziness are uncommon, with a prevalence of 0.4–15% [3,4,5]. Although it is well known that children seldom complain about vertigo and dizziness, many concerns about the precision of diagnosis emerge from the brief duration of the symptoms, difficulties in expressing the problem, underestimation of the problem with the parents, and the scarcity of studies on the topic. Moreover, other issues challenging the accuracy of these findings may include heterogeneity of the study designs, of data gathering, source and analysis, and of inclusion and exclusion criteria [6].

Causes of central vertigo and dizziness include disorders of the vestibular nuclei of the ponto-medullary brainstem and of the pathways that connect vestibular nuclei to the cerebellum, brainstem, thalamus, and cortex. On the other hand, peripheral vertigo includes damage to the labyrinth and vestibular nerve [7]. Common causes of central and peripheral vertigo in children include congenital defects, heritable diseases, and acquired diseases; in addition, central and peripheral vertigo in children can be caused as a result of head trauma, which can appear in forms of acute or chronic, permanent, or paroxysmal balance disorder. Hearing loss, ataxia, anorexia, vomiting, abdominal pain, double vision, nystagmus, and many other signs or symptoms could be associated with balance disorder and lead to diagnosis, which is obtained through a detailed neurological, otological, and audio vestibular clinical examination complemented by instrumental investigations when appropriate [8]. Among the possible diagnosis, common balance disorders in children include vestibular migraine of childhood (VMC), recurrent vertigo of childhood (RVC), vestibular neuritis (VN) and, with less incidence, motion sickness (MS), persistent perceptual postural dizziness (PPPD), benign paroxysmal positional vertigo (BPPV), and Menière disease (MD), which, conversely, are more common in the adult population [8,9]. The purpose of this review is to provide a comprehensive overview of the available medical treatments for the most frequent balance disorders in children.

## 2. Methods

We searched the PubMed, Embase, and Cochrane library databases for articles published until 10 March 2022 in English. The secondary search included articles from reference lists, identified by the primary search. Records were first screened by title/abstract and then full-text articles were retrieved for eligibility evaluation. The searches combined a range of key terms (“Pediatric” AND “Childhood” AND “dizziness” OR “vertigo” AND “vestibular”). Duplicate manuscripts were removed after exporting references to the Mendeley reference management software (https://www.mendeley.com, accessed on 13 December 2021).

## 3. Results

### 3.1. Vestibular Migraine and Recurrent Vertigo of Childhood

VMC, probable VMC and RVC (previously Benign Paroxysmal Vertigo of Childhood—BPVC) [10], are the most common cause of vertigo and dizziness in children. It is estimated that the incidence of BPVC is about 3% [11]; the disease begins before the age of four and spontaneously resolves at the age of 10 [12]. The disease is characterized primarily by the recurring of short spontaneous vertigo attacks without connection to position, which may lead to vomiting, paleness, postural imbalance, and ataxia, with or without nystagmus. On the other hand, the term “vestibular migraine” (VM) is widely accepted and is the most widely used name for vestibular symptoms associated with migraines. VM is generally considered the most common cause of vertigo in adults and children, although the relative frequency in the latter population is higher [13,14]. Since their first proposal in 2004, the diagnostic criteria for vestibular migraines have been well established and validated by the International Headache Society (IHS) and the Committee for the International Classification of Vestibular Disorders (ICVD) of the Bárány Society [15,16,17]. However, the diagnostic criteria for VM are not validated in children, despite these being applied as well, considering the absence of any age limit of application. Nowadays, the Classification Committee of Vestibular Disorders of the Bárány Society and the IHS updated the guidelines, separating the entities by age in VM and VMC, and tailoring specific diagnostic criteria for children [10]. The onset of both diseases is often not recognized until the child is old enough to describe their symptoms properly; thus, the exact incidence of BPVC/RVC and VMC is not well known. It is questionable whether BPVC/RVC and VMC are separate entities or part of the same spectrum, considering that several investigations have found evidence of progression from the former to the latter (Table 1, Table 2 and Table 3) [18,19]. Moreover, a significant percentage of children with vertigo satisfies diagnostic criteria for both diseases; thus, there is a not negligible likelihood that patients with episodic vertigo could be diagnosed with both BPVC/RVC and VMC (Table 1, Table 2 and Table 3). Guidelines for diagnosis and treatment of RVC and VMC have not yet been established and there are only few suitable treatments for children (Table 1, Table 2 and Table 3). In our experience, a prophylaxis is indicated when attacks are more than three per month or if symptoms are severe. On the other hand, as medical treatment data are limited in childhood, often most experiences are on headache and not always on vertigo. For these reasons, we believe that the approach to VMC and RVC must go through a decision-making process shared with parents as well. Considering the lack of evidence, the prophylactic treatment of VMC and RVC relies on the same medications used for migraine headache [10,13,20,21], therefore, there are no specific drugs for migraines. Few data have been published concerning the efficacy of antimigraine drugs on vestibular symptoms of VMC (Table 1, Table 2 and Table 3). Randomized controlled clinical trials of adult VM treatment are rare, while those involving children do not exist. A retrospective analysis of 28 patients (9–18 years) showed that several drugs used as a prophylactic treatment reduce vestibular symptoms, i.e., tricyclic antidepressant (88% *n* = 8), cyproheptadine (86%; *n* = 7), topiramate (80%; *n* = 5), triptans (80%; *n* = 10), and gabapentin (25%; *n* = 4). Vestibular suppressants (i.e., meclizine, prochlorperazine, promethazine, ondansetron, and scopolamine) were administered as acute treatment in seven subjects without a clinical improvement [22]. In 14 children affected by basilar-type migraines, topiramate (25–100 mg) significantly reduced the frequency of migraines compared to baseline, even if the absence of a placebo control group impairs the validity of these results [23,24,25]. Concerning RVC, it is self-limiting but the persistence of symptoms could require antiemetics [26].

### 3.2. Vestibular Neuritis

VN is not the most common diagnoses of vertigo in children [5,8,52]. The specific etiology of VN remains to be addressed, however, it is associated with rhino-pharyngeal and herpes simplex virus infections [53], or with autoimmune mechanisms [54]. In pediatric patients, vestibular neuritis arises during a viral infection as serious rotatory vertigo, associated with nausea and emesis. The clinical examination shows clear evidence of isolated peripheral vestibular deficit: when seated on the floor, the child tends to fall repeatedly to the same side, close the eyes, and shows nystagmus. Otologic and neurologic examinations show unilateral vestibular impairment without neurologic involvement. In childhood, VN is rapidly compensated and the risk of incomplete resolution is rare; therefore, the treatment is symptomatic [55,56].

In a retrospective study, 2 of 11 children with VN received oral prednisone one week after the beginning of the symptoms and vestibular rehabilitation, resulting in a complete clinical improvement [56]. However, the small sample size and the study design represent a limitation to this study [56]. In this regard, it is important to note that the results provided by a Cochrane systematic review, considering adults with VN, showed insufficient evidence to support corticosteroid use [57]. Symptomatic treatments, such as anti-nausea and antiemesis drugs, are available therapeutic options [58]. Dimenhydrinate (1–2 mg/kg) is one of the more viable alternatives, although a short-term usage (3 days maximum) is suggested [50].

### 3.3. Paroxysmal Positional Benign Vertigo (BPPV)

Despite being the most common cause of vertigo in adults, BPPV is rarely diagnosed in children, except in patients with head trauma. BPPV is a disorder of the vestibular labyrinth that causes episodes of brief and severe vertigo related to positional changes of the head, associated with canal-related nystagmus due to the dislocation of otoliths crystals from the utricle to the semicircular canals [59]. The oto-neurological examination shows positional-evoked nystagmus. Many physical maneuvers have been described as the first-line therapy of the disease, such as Semont and Epley [59,60]. To date, there are no recommendations for the treatment of BPPV in children and clinicians should not routinely use drugs to manage BPPV [61]. The few available papers on this topic are focused on maneuvers, rather than drugs, as the main therapeutic option for BPPV [27,59,62,63,64]. Balzanelli et al. [65] suggested that a pharmacological treatment could be used, without reporting any evidence about specific compounds. Residual dizziness after BPPV can be treated with the supplementation of polyphenol compounds able to reduce subjective symptoms and improve instability earlier [66]. However, Wipperman stated that the use of drugs may be futile, even dangerous, since their efficacy has not been strongly evaluated and, when examined, seemed to be lower than other therapies. Moreover, drugs such as antihistamines or benzodiazepines may impair vestibular compensation [67].

### 3.4. Menière Disease

MD rarely affects pediatric patients, showing a greater prevalence in the age between 8 and 10 years. Considering the many difficulties related to the communication with children, it is possible that they cannot clearly describe its symptoms hindering the making of a correct diagnosis. Moreover, aural symptoms in children arise over a long period of time, thus, retarding the diagnosis of disease, even when symptoms can be clearly described. Several drugs could be used in pediatric patients with MD, e.g., isosorbide dinitrate, dimenhydrinate, diuretics, hydrocortisone, and flunarizine [68,69,70,71,72].

Nevertheless, none of the treatments addressed the question about the best therapy in terms of safety and efficacy [73]. Meyerhoff et al. described the role of anti-cholinergic drugs and vestibular depressants [74]. There are anecdotal reports related to the use of dihydroergotamine and ibudilast [75]. Other authors wrote about the role of nutrients (such as *Ginkgo Biloba*) and trimetazidine [76]. However, some cases are self-resolving without the administration of any treatment [77].

#### 3.4.1. Isosorbide Dinitrate

Isosorbide dinitrate (ID) has a vasodilator and hypotensive effect [78]. MD pathogenesis is related to endolymphatic hydrops and ID reduces the endolymphatic pressure, improving the circulation in the stria vascularis [60]. Even if ID is not well reported in children [78], some authors reported its use in pediatric patients [73,75,79]. The most common adverse drug reactions (ADRs) during ID treatment are represented by headache, orthostatic hypotension, reflex tachycardia, nausea, abdominal discomfort, and weakness [78,80].

#### 3.4.2. Diuretics

The evidence supporting the use of diuretics in MD is low, even if their role in the reduction in endolymphatic hydrops is reported. [60]. In juvenile MD, Filipo and Barbara [81] reported an improvement in symptoms after treatment with diuretics, particularly chlortalidone. Brantberg et al. [82] reported the use of thiazides (hydrochlorothiazide and bendroflumethiazide) in children (4–7 years). Meyerhoff et al. [74] stated that, in absence of a specific etiology, the use of diuretics alongside other compounds such as vestibular depressants, anti-cholinergics, vasodilators, and anti-histamines is a viable option. Wang et al. [73] assessed the importance of diuretics in the treatment of pediatric MD, but stressed that none of the available therapies seem to be more effective than others. Choung et al. [76] described the use of spironolactone and hydrochlorothiazide, showing a good recovery in hearing function.

However, hydrochlorothiazide (and other thiazides) may induce several ADRs, e.g., impaired glucose tolerance, increase in cholesterol, triglycerides, and calcium [83,84,85]. Hydrochlorothiazide increases the risk of skin cancer [86] while spironolactone induces hyponatremia, hyperkalaemia, hypovolemia, metabolic acidosis, vocal alterations, gastrointestinal symptoms, and sexual/hormonal changes (e.g., gynecomastia and hirsutism) [87,88].

#### 3.4.3. Other Therapeutic Options

The antagonist on the H_1_ (histamine) receptor [89] is involved in vestibular compensation [90]. The clinical use of these drugs for MD is common in adults [60,91], while little evidence exists regarding children. Wang et al. [73] assessed that dimenhydrinate can be used in children. However, dimenhydrinate showed many important adverse events, such as somnolence, cholinergic effects (dry mouth, blurred vision and mydriasis, and urinary retention), headache, and gastrointestinal symptoms [92].

Flunarizine, a calcium antagonist, is indicated for the prophylaxis of migraines and the treatment of vertigo and seizures. It inhibits vascular smooth muscle contraction, protecting endothelial cells, red blood cells, and brain cells. Moreover, flunarizine has showed vestibular depressing, antihistamines, and anticonvulsant effects [37].

In 24 children with MD, flunarizine (2.5–5.0 mg/day) induces an improvement in vertigo. During the follow-up, 55% of patients described a recurrence in clinical manifestations, with an interval of 3.7 ± 2.9 years [73]. In Italy, flunarizine is contraindicated in children [93]. ADRs include rhinitis, increased appetite and weight gain, drowsiness/insomnia, depression, gastrointestinal symptoms, and myalgia [37,94]. Hydrocortisone can be also used even if the mechanism in MD has not been explained [60]; Akagi et al. [75] reported that after being treated with hydrocortisone, ibudilast, mecobalamin, and adenosine triphosphate disodium (ATP), children showed a good clinical improvement. Oral corticosteroids can be alco used in MD treatment [95].

### 3.5. Persistent Postural-Perceptual Dizziness in Children and Adolescents

Recently, a new syndrome called “persistent postural-perceptual dizziness” (PPPD) has been described. PPPD is a functional neurological disorder associated with non-vertiginous dizziness and a sense of imbalance exacerbated by position changes, standing, turning, and visual flow. It is precipitated by an event that causes dizziness, vertigo, or imbalance, such as a peripheral or central vestibular disorders, psychological distress, or other illnesses. Patients may also report a fear of falling, and often present with constant low-level dizziness at baseline, as well as frequent flares. Symptoms are commonly worsened by large, open spaces, and high-flow visual settings, such as grocery stores or shopping malls. Diagnostic criteria [66,96], as well as treatment in pediatrics, still lack well-established clinical guidelines. A retrospective study on 53 pediatric patients (14.6 ± 3.2 years) with PPPD [97] evaluated the efficacy of CBT or biofeedback therapy, physical therapy, and selective serotonin reuptake inhibitor (SSRI) or serotonin noradrenaline reuptake inhibitor (SNRI). In this study, the authors documented that among the 18 patients who reported complete symptoms resolution, seven underwent triple therapy with CBT, PT, and SSRI/SNRI.

In a systematic review, Solmi et al. [98] reported that in children and adolescents, antidepressants induce nausea/vomiting, weight gain, sedation, extrapyramidal side effects, diarrhea, headache, and anorexia. Escitalopram and fluoxetine were associated with a minor number of dropouts and increased safety, whereas venlafaxine was more probably related to ADR. There are no trials for SSRI and SNRI in children with PPPD: we summarized their current clinical indications in children in Table 4.

### 3.6. Motion Sickness

MS is an ever-increasing phenomenon due to the increasing travel and screen time in the modern era. The pathophysiology of MS is related to a sensory conflict between vestibular, visual, and proprioceptive inputs [99,100]. The clinical characteristics of the disease include emesis, nausea, vertigo and dizziness, asthenia, paleness, and headache. Triggering stimuli of these distressing symptoms could be travelling via boat, plane, and car. Although extensive literature exists about MS in adults (prevalence 13.4%), the same disease in children has not be well characterized (prevalence 43%) [101,102]. In addition, the treatment for MS has not been well investigated (Table 5, Table 6 and Table 7).

Cyproheptadine is an antihistaminic drug with efficacy as a prophylactic treatment for migraines in children due to its anticholinergic and anti-serotoninergic properties [36]. In a retrospective study, Lipson et al. [103] enrolled 23 patients (0–15 years), and were treated with cyproheptadine, meclizine, ondansetron, acupressure, tympanostomy tube placement, change of car seat position, and vestibular rehabilitation. The authors documented that cyproheptadine is a promising treatment [103] (Table 6).

Meclizine, a first-generation antihistamine drug with central anticholinergic action [104], is used to treat symptoms of MS and vertigo. It inhibits the signals through histamine neurotransmission from the nucleus of the solitary tract and the vestibular nuclei to the chemoreceptor trigger zone and vomiting center, located in the medulla [105]. Meclizine is approved by FDA for children over 12 years at the dosage of 25 to 50 mg, one hour before embarkation or triggering situation; additional administration is possible once every 24 h [106]. Although the drug appears to have a good effect against MS through vestibular modulation, the effectiveness of meclizine has been questioned [107,108]. Common ADRs include drowsiness, urinary retention, dry mouth, headache, fatigue, and vomiting [109,110]. Caution is needed in patients with glaucoma due to its anticholinergic properties [111] (Table 5).

Ondansetron is a selective 5-HT3 serotonin-receptor antagonist used in the prevention of nausea and vomiting induced by surgery, chemotherapy, or radiotherapy. It has a minimal effect on nausea and vomiting caused by motion [112,113]. Ondansetron acts both centrally and peripherally. Central effects are mediated by the antagonism of 5HT-3 serotonin receptors in the area postrema located on the fourth ventricle floor, containing the “chemoreceptor trigger zone.” The peripheral action is on 5-HT3 receptors in the terminals of the vagus nerve, responsible for nausea and vomiting triggered by the gastrointestinal tract. Pediatric dosing is 0.15 mg/kg with a maximum of 16 mg per dose; however, only few data about the pediatric population are available [114,115] (Table 5).

Scopolamine is one of the most prescribed drugs for the treatment or prevention of MS. It inhibits G protein-coupled post-ganglionic muscarinic receptors for acetylcholine and acts as a non-selective muscarinic antagonist, producing both peripheral antimuscarinic properties and central sedative, antiemetic, and amnestic effects [116]. The vomiting center in the medulla oblongata contains a high amount of M1, H1, NK1, and 5-HT3 receptors. Scopolamine exerts its action by primarily affecting the M1 receptor, even if the H1 receptor could be also involved [117,118]. This drug has two major contraindications: allergy to belladonna alkaloids and angle-closure glaucoma [119]. Although scopolamine has been used for decades in the treatment and prevention of MS, there are no studies on its efficacy and/or safety. Furthermore, it is not indicated in children. Doses for adults are typically 0.3 to 0.6 mg per day, while smaller doses of about 0.006 mg/kg are given to children. No studies are available on the therapeutic efficacy of scopolamine in the management of established symptoms of MS [120] (Table 5).

Promethazine is used to treat allergic conditions, nausea, and vomiting, and as prophylactic therapy for MS. Promethazine is a phenothiazine derivative with antidopaminergic, antihistamine, and anticholinergic properties [121,122]. The dosage of tablets, solutions, and suppository is generally 12.5 mg to 50 mg. There is also a syrup form of 6.25 mg/5 mL [123,124]. In pediatrics, dosing adjustments are needed in function of both patient weight and clinical indication. It is most effective when given 30 min or 1 h before the triggering event [125]. ADRs include sedation, confusion, and disorientation. Due to its anticholinergic properties, promethazine may cause blurred vision, xerostomia, dry nasal passages, dilated pupils, constipation, and urinary retention. Due to its antidopaminergic properties, it causes extrapyramidal symptoms, including pseudo-parkinsonism, acute dystonia, akathisia, and tardive dyskinesia [126]. It is contraindicated in children under two years of age due to the risk of potentially fatal respiratory depression [127,128]; however, some countries allow the use of this drug in children ranging from 6 months to 2 years [129] with careful management of intramuscular dosage and in emergency situations only (Table 5).

Cinnarizine is a piperazine derivative with antihistaminic, antiserotonergic, antidopaminergic, and calcium channel-blocking properties. It is currently used for the treatment of nausea, vomiting, vertigo, and the prevention and treatment of MS [130]. It reduces both frequency and severity of migraines in children, particularly when it is associated with vestibular symptoms [32,131,132,133]. Conflicting results were found in two studies comparing the efficacy of scopolamine with cinnarizine. Pingree et al. found that scopolamine was better than cinnarizine in preventing symptoms, while Hargreaves et al. have shown that cinnarizine is superior to scopolamine [134,135]. The dose in adults varies from 30 to 75 mg, 2 h before the beginning of the trip or triggering situation, repeating lower doses of 15 to 50 mg every 8 h. The cinnarizine-recommended dosage for children between 5 and 12 years old corresponds to half of the dose used in adults [136]. Macnair et al. [137] evaluated, in 79 children (6–12 years) susceptible to car sickness, the prophylactic efficacy and safety of cinnarizine. Each child received cinnarizine (15 mg) 2 h before a long car journey, and half a dose every 8 h, if required. A total of 81% of enrolled children considered the preparation as either ‘good’ or ‘excellent’, and 69% as ‘better’ or ‘much better’. However, 4% of the children vomited and only 14% felt sleepy or drowsy (Table 6).

Flunarizine is the difluorinated derivative of cinnarizine with antihistamine and calcium-antagonist properties. The drug is used in the treatment of vertigo and migraines. Its efficacy in the prevention of MS remains to be addressed [138,139] (Table 6).

Esposito et al. [49] evaluated, in 254 prepubertal children, the efficacy and safety of the Griffonia simplicifolia/Magnesium (50 and 200 mg, respectively) as a prophylactic treatment for MS. The Griffonia/magnesium complex was orally administered for 3 months to group A, while no therapy was administered to group B. At the end of the study, group A showed a significantly lower prevalence (36% vs. 73%; *p* < 0.001) and severity (evaluated through the visual analogical scale) of MS symptoms than group B (2.59–0.14 vs. 6.91–2.08; *p* < 0.001). No ADRs were recorded, except sporadic diarrhea and stomach ache during the first 2 days of treatment (3.14% of the subjects in group A). These results suggest that Griffonia/magnesium complex is a potential treatment for MS [49] (Table 7).

**Table 5 children-09-00584-t005:** Rationale, current clinical indications, and dosages of peripheral vestibular vertigo drugs in children (indication to use).

	Mechanism(s) of Action	Current Clinical Indications	Dosage Suggested	Route of Administration
**Anticholinergics**				
Scopolamine	Non-selective muscarinic blocker [140]	Motion sickness (avoid in children under 10 years of age) [141].	1 mg (TD) [141] 0.006 mg/kg (IM) [141] dose, repeat every 6–8 h.	IM, IV, TD, OS, nasal spray [141]
**Antihistamines**				
Dimenhydrinate	Antagonist of H1 receptor [142]	Motion sickness. No clinical trials in VN and MD, although suggested by some authors [50,73,141].	2–6 years: 25–75 mg (OS). 7–12 years: 25–150 mg (OS) or 1–2 chewing gum of 25 mg [92]. 1.25 mg/kg of body weight or 37.5 mg/m 2 of body surface area four times daily (IM, maximum 300 mg) [143]. 1–2 mg/kg in VN [50].	IM, OS (cps. chewing gum) [92,143]
Meclizine	Antagonist of H1 receptor [144]	Motion sickness (use carefully under 12 years, not available in Italy) [103,141,145].	25–50 mg daily in children over 12 years [106,146,147].	OS [146]
Promethazine	Antagonist of H1 receptor. Antidopaminergic and anticholinergic properties [121,122]	Motion sickness (in the USA, off-label in Italy) [148].	12.5 mg to 50 mg. There is also a syrup form of 6.25 mg/5 mL (OS) [123,124]. In pediatrics, dosing adjustments are needed in function of the patient weight and the indication. For children, promethazine hydrochloride tablets, syrup, or rectal suppositories, 12.5 to 25 mg, twice daily, may be administered [149]. Contraindicated under 2 years of age [127,128]. 2–5 years: 5–7.5 mg. 5–10 years: 7.5–12.5 mg. 25 mg in general population (IM) [129].	OS, IM [129]

IM, intramuscular; IV, intravenous; MD, Meniere’s Disease; OS, oral; TD, transdermal; VN, vestibular neuritis.

**Table 6 children-09-00584-t006:** Rationale, current clinical indications, and dosages of peripheral vestibular vertigo drugs in children (off-label compounds).

	Mechanism(s) of Action	Current Clinical Indications	Dosage Suggested	Route of Administration
**Antihistamines**				
Cinnarizine	Antihistaminic, antiserotonergic, antidopaminergic, and calcium channel-blocking activities [130].	Motion sickness (more properly in balance disorders, not available in the USA) [130,150].	30–75 mg 2 h before the start of the trip, repeating lower doses of 15 to 50 mg every 8 h (in adults). No information of possible dosage in 12–18 years. Children 5–12 years: 15–25 mg 2 h before departure; repeated doses of 7.5–15 mg if necessary [136] ^a^.	OS
Cyproheptadine	Antagonist of H1 receptor [34]. Serotonin antagonist [35] and anticholinergic effect [36].	Motion sickness [103]	Not specified in dedicated paper. *SmPC dosages below.* Adults: 4–20 mg daily, 2–6 years: 2 mg twice or thrice daily (max 12 mg), 7–14 years: 4 mg thrice daily (max 16 mg) [151].	OS
Flunarizine	Antagonist of H1 receptor and calcium antagonist [37].	MD	2.5–5.0 mg daily in one clinical trial for MD (24 children <15 of age) [73].	OS

MD, Meniere’s Disease; OS, oral; TD, transdermal; VN, vestibular neuritis. ^a^ = contraindicated in children Italy and the USA.

**Table 7 children-09-00584-t007:** Rationale, current clinical indications, and dosages of peripheral vestibular vertigo drugs in children (potentially useful drugs, with no trials in children).

	Mechanism(s) of Action	Current Clinical Indications	Dosage Suggested	Route of Administration
**Diuretics**				
Bendroflumethiazide	Inhibition of sodium chloride co-transporter in the distal convoluted tubule [83,152].	MD [82]	1.25 mg daily in a case report [82] (6-year-old child).	OS
Hydrochlorothiazide	Inhibition of sodium chloride co-transporter in the distal convoluted tubule [83,152].	MD [76,82]	6.25 mg [82] (7-year-old child) ^a^.	OS [83]
Spironolactone	Mineralocorticoid receptor antagonist [153].	MD [76]	Not specified in dedicated papers ^b^.	OS [87,153]
**Other drugs**				
Griffonia simplicifolia/Mg	Serotoninergic action [47,48,49]	Motion sickness [49]	Pediatric dosing data are not available. 50/200 mg twice a day, respectively, in adults [49].	OS
Hydrocortisone	Anti-inflammatory effect, acting on gene transcription [154]	MD (low evidence) [60,73,75] and VN [155]	Pediatric dosing data are not available. Further studies are needed in population under 18 years. Oral formulations are available for adults in 5 mg and 20 mg doses. (20–40 mg maintenance dose) [154]. Dosage is generally based on weight [156].	OS
Isosorbide dinitrate	Vasodilator and hypotensive effect [78]	MD [73,75,79]	Pediatric dosing data are not available; 5–80 mg daily in adults formulation [78].	OS
Methylprednisolone	Anti-inflammatory effect, acting on gene transcription [157]	VN [155] and MD (low evidence) [60]	Pediatric dosing data are not available; 4–48 mg in general population. Dosage is generally based on weight [156,158].	OS [56]
Ondansetron	5HT3 antagonist [83,159]	Motion sickness [103]	Pediatric dosing data are not available; 5 mg/m^2^ or maximum three doses of 0.1–0.15 mg/kg every 4 h (IV, max intravenous dose 4–8 mg). BSA <0.6 m^2^ (≤10 kg): 2 mg twice daily, BSA ≥ 0.6 m^2^ (>10 kg): 4 mg twice daily (OS, max daily dose 32 mg) [83].	IV, OS [83]

BSA, body surface area; IM, intramuscular; IV, intravenous; MD, Meniere’s Disease; OS, oral; VN, vestibular neuritis. ^a^ = contraindicated in children Italy and the USA, ^b^ = contraindicated in children in the USA.

## 4. Conclusions

Vertigo in children is a quite neglected topic and only a few drugs, such as antihistamines (promethazine, meclizine) or anticholinergics (scopolamine, prochlorperazine), can be used [60,160]; the choice of drug used is related to the age of the patients. For example, prochlorperazine should not be used in children younger than 2 years of age and dose adjustments may be necessary based on age [161]. Scopolamine, meclizine, and promethazine need to be managed in the same way (see Section 3.6). The absence of specific pediatric guidelines for many of these pathologies is a crucial issue.

The management of these clinical pictures must, therefore, be guided by the etiological diagnosis, which, often, does not require pharmacological treatment. In some cases, a vigilant observation or a rehabilitative approach can be the better choice. On basis of available experience, in cases where a pharmacological treatment is indicated, it should be integrated with rehabilitation or cognitive-behavioral therapies.

Several drugs have been tested in some experimental studies and are evaluated by physicians for the effect that they have shown in adult patients, taking into account their personal experience. There are great differences in age prescription by each country, creating serious difficulties to practitioners. Moreover, the majority of the trials performed for the drugs possibly used in vestibular migraines refer generally to migraine children/adolescent courts. In our opinion, the recruitment of courts focused on vestibular migraines could give more specific information. Although their possible use is anecdotical and based on a unique paper, we described SNRI and SSRI, summarizing their current indication. No guidelines or clinical trials indicate antidepressant use in PPPD, so their use is far from clinical practice and appropriateness. In this situation, clinical trials must be performed in children with vertigo to obtain clinical data useful to treat this clinical situation that is related to chronic pain and a decrease in the quality of life.

## Figures and Tables

**Table 1 children-09-00584-t001:** Drugs used or evaluated in vestibular migraine and recurrent vertigo of childhood (*off-label*, part 1).

	Mechanism(s) of Action	Dosage Suggested	Route of Administration
**Antiepileptics**			
Levetiracetam	Not completely clear. It seems to act on intraneuronal calcium levels, inhibiting N-type calcium currents and lowering calcium release. Modulation of GABA and glycine gated currents. Binding to synaptic vesicle protein 2A.	20–40 mg/kg/day [27,28,29] in children aged 4–17 in a clinical trial for migraine prophylaxis (*off-label*) Not approved in children under 12 years old (the USA, seizures). Not approved under 16 years in Italy.	OS
Topiramate	Reduction in voltage-gated sodium channel currents. Activation of potassium and GABA_A_ receptor currents. Blocking of AMPA/kainate receptors. Weak inhibitor of carbonic anhydrase.	No trials on vestibular migraine patients but used for migraine prophylaxis (*on-label*). 1–4 mg/kg/day in two doses [21,27,30] titrated slowly in 8–12 weeks. It may be given in children ≥2 years, but is approved for migraines in patients ≥12 years in the USA. Children ≥12 years: 50 mg BID with a gradual titration [31].	OS
Valproic acid	Possible increase in GABA levels	Migraine prophylaxis (*off-label*) 10–30 mg/kg/day [27,30,32]. Risk of serious adverse events in children <3 years: use only if there is urgent need and in monotherapy.	OS
**Antidepressants**			
Amitriptyline	Anticholinergic and antiadrenergic properties. Inhibition of norepinephrine and serotonin uptake.	No placebo-controlled trial, but some data have been collected. Used in clinical practice for migraine prophylaxis (*off-label*). A total of 0.5–1 mg/kg/day. Because of its side effects, slow titration in 8–12 weeks to the goal dose of 1 mg/kg/day (dose increase in 0.25 mg/kg/day every two weeks) [27,30]. Not recommended in patients under 12 of age [33] ^a^.	OS
**Antihistamines**			
Cyproheptadine	Antagonist of H1 receptor [34]. Serotonin antagonist [35] and anticholinergic effect [36].	Migraine prophylaxis (*off-label*). A total of 0.2–1.5 mg/kg/day (0.2 mg/kg/day is considered the most common dosage) mainly in children under 6 years of age [30]. Use in children ≥2 years only.	OS
Flunarizine	Antagonist of H1 receptor and calcium antagonist [37].	Migraine prophylaxis (*off-label)*—5–10 mg/day [27,30,38] ^b^.	OS

If not differently specified, data reported can be found in Summary of Product Characteristics (SmPC). AMPA, α-amino-3-hydroxy-5-methyl-4-isoxazolepropionic acid; GABA, gamma-aminobutyric acid; OS, oral. ^a^ = contraindicated in children under 18 years in Italy. ^b^ = contraindicated in children under 18 years in Italy, not available in the USA.

**Table 2 children-09-00584-t002:** Drugs used or evaluated for vestibular migraine and recurrent vertigo of childhood (off-label, part 2).

	Mechanism(s) of Action	Dosage Suggested	Route of Administration
**β-blockers**			
Propranolol	Non-selective, beta-adrenergic receptor-blocking agent	Migraine prophylaxis (*off-label*) in children aged 3–15 years—1–4 mg/kg/day [21,27,30,39,40] ^a^.	OS
**Triptans**			
Almotriptan	Agonist of 5HT1D receptor	Tested in adolescents 12–17 years at 12.5 mg [41]. Approved for migraine treatment in patients of 12–17 years with a history of migraines with or without aura, and who have migraine attacks that usually last 4 h or more: dosage 6.25–12.5 mg ^b^.	OS
Rizatriptan	Agonist of 5HT1B and 5HT1D receptors	Migraine treatment (*on-label*, the USA). <40 kg: 5 mg. ≥40 kg: 10 mg [42,43] in patients 6–17 years old (OS) ^b^.	OS
Zolmitriptan	Agonist of 5HT1B and 5HT1D receptors. It has also a minor action on 5HT1A	Migraine treatment A 2.5 mg (OS) dosage showed good results in children of 6–13 years [44] ^a^ in clinical trials (*off-label*).	OS, NS
		A total of 5 mg (NS) in patients of 12–17 years [45] ^b^ is approved for migraine treatment (*on-label*, the USA).	
**Other drugs**			
Coenzyme Q10	Antioxidant action. It also favors mitochondria physiology [21].	Migraine prophylaxis:100 mg in children ≥3 years [46].	OS
Magnesium aspartate	Serotoninergic action [47,48,49]	Migraine prophylaxis: 50/200 mg twice a day, respectively [49]. A total of 200–400 mg or 9 mg/kg divided three times daily in children of 3–17 years [30,50].	OS

If not differently specified, data reported can be found in SmPC, AMPA, α-amino-3-hydroxy-5-methyl-4-isoxazolepropionic acid; BSA, body surface area; GABA, gamma-aminobutyric acid; IM, intramuscular; IV, intravenous; OS, oral; TD, transdermal. ^a^ = contraindicated in children under 18 years in Italy and the USA, ^b^ = contraindicated in children under 18 years in Italy.

**Table 3 children-09-00584-t003:** Drugs used or evaluated in vestibular migraine and recurrent vertigo of childhood (no trials).

	Mechanism(s) of Action	Dosage Suggested	Route of Administration
**β-blockers**			
Metoprolol	Selective β1 receptor blocker	0.5–1 mg/kg/day [50] for migraine prophylaxis. Safety and effectiveness of metoprolol succinate have not been established in patients <6 years of age [51] ^a^.	OS
**Other drugs**			
Riboflavin	It favors mitochondria energy cycle [21]	Migraine prophylaxis: 200–400 mg [21] in children/adolescents 9–19 years in a retrospective study.	OS

If not differently specified, data reported can be found in SmPC, AMPA, α-amino-3-hydroxy-5-methyl-4-isoxazolepropionic acid; BSA, body surface area; GABA, gamma-aminobutyric acid; IM, intramuscular; IV, intravenous; OS, oral; TD, transdermal. ^a^ = contraindicated in children under 18 years in Italy and the USA.

**Table 4 children-09-00584-t004:** SSRI and SNRI antidepressants (no clinical trials available for PPPD).

	Mechanism(s) of Action	Age	Route of Administration
**SNRI ***	Inhibition of both serotonin and norepinephrine reuptake	-	-
Duloxetine	-	Children ≥7 years in generalized anxiety disorder ^a^	OS
Venlafaxine	-	Not approved in children/adolescents under 18 years	OS
**SSRI ***	Inhibition of serotonin reuptake	-	-
Citalopram	-	Not approved in children/adolescents under 18 years	OS
Escitalopram	-	≥12 years for depression ^a^	OS
Fluoxetine	-	≥8 years for depression	OS
Fluvoxamine	-	≥8 years for OCD ^a^	OS
Paroxetine	-	Not approved in children/adolescents under 18 years	OS
Sertraline	-	≥ 6 years for OCD	OS

If not differently specified, data reported can be found in SmPC. * = Dosages in PPPD are not yet available. OCD, obsessive compulsive disorder; OS, oral; SNRI, serotonin and norepinephrine reuptake inhibitors; SSRI. ^a^ = contraindicated in children under 18 years in Italy.

## Data Availability

Not applicable.

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
