# Peer review of "The Pharmacological Treatment of Pediatric Vertigo"

_children, 2022, doi:10.3390/children9050584_

Round 1

Reviewer 1 Report

There are some relevant issues that must be clarified (see comments to lines 146-147, and 165).

Introduction: the definitions of Bárány Society to define vertigo, dizziness and unsteadiness are preferred.

Line 47: the word “central” seems to have been omitted when referring to central vertigo.

Line 75 2.1. “Vestibular migraine and recurrent vertigo of childhood”. This and the others subheading should not be included as part of the methods section.

Line 83: please, include the reference to the criteria of vestibular migraine (Lempert et al, 2012) as VM is not only “vestibular symptoms associated with migraine”.

Line 101: reference 10 is a diagnostic guideline as you have mentioned in line 92.

Line 110: you can distinguish between preventive treatment and acute attack treatment.

Line 120: The abbreviation SmPC is not previously established along the text.

Table 3 and others: “die” is not the international symbol of day.

Vestibular neuritis

Line 140: VN is NOT the most common diagnoses of vertigo in children, as you have stated at lines 76-77.

Line 146-147: the sentence “Otologic and neurologic examinations did not show vestibular or neurologic impairment” is wrong. Unilateral vestibular impairment, as evidenced by head impulse test, is always present. Please, review.

BPPV:

Line 165: Equally, the sentence “Both otologic and neurologic examinations are normal” is wrong.  The oto-neurological examination shows an important sign in BPPV. Please, review.

Meniere´s disease

Line 180: What exactly does this mean “showing a greater prevalence in children”?

Line 191: trimetazidine is not recommended by the European Medicines Agency in this indication.

Line 203: The evidence supporting the use of diuretics in MD is low (Crowson et al., 2016; Rosenbaum and Winter, 2018).

Other comments:

- Dopamine antagonists such as metoclopramide and domperidone are antiemetics that are not discussed in the manuscript. Certainly, metoclopramide has an increased risk of extrapyramidal symptoms in children and the European Medicine Agency has restricted its use; on the other way, domperidone is associated with an increased risk of developing a prolongation of the QT interval in the pediatric age. This reviewer consider that it is worth noting these riks because of the extended use of these antiemetics.

Tables:

- The left/first column is unclear in most of the tables because the name of the pharmacological group (antiepileptics, betablockers, triptans. etc.) appears with the same font style than the drugs (Levetiracetam, topiramate, etc.)

Reviewer 2 Report

Very nice piece, and very important!

Introduction
- Do you think that vertigo and dizziness is uncommon in children or underdiagnosed? I see a lot of children where I diagnose vertigo but they haven't complained of dizziness (due to impairment in expression for a variety of reasons)
- It may be helpful to also mention that vertigo can be a result of head trauma in pediatric patients

Results?
- The methods section is section 2, but it seems that you start discussing results as 2.1?
- Under 2.1, you mention that attacks for Recurrent Vertigo of Childhood are > 10 minutes. Is this a typo? A range may be more helpful. 
- Discussing that the time frame reported by the patient or family may not be as reliable (underestimated or overestimated) may lead to less confusion in clinical practice for the reader
- Amitriptyline + cognitive behavioral therapy is recommended for patients 10-17 years of age according to pediatric migraine prevention guidelines, whereas in your table it mentions that it is not recommended under the age of 12
- Triptans- I had difficulty looking at your table regarding whether there was data for vertigo treatment? 
- Magnesium- works on NMDA for migraine, not sure about proposed mechanisms in vertigo, however
- The table was a bit crowded, which may have made it more difficult to read, and beta blockers were separated into 2 different places
- For the other types of vertigo, there was a detailed discussion on each treatment with contemplation of evidence, but not for 2.1. I think a discussion of the studies (at least a little bit) rather than just the table is warranted.

Round 2

Reviewer 1 Report

This reviewer appreciate the new version of the manuscript. My comments have been fully addressed. Thank you.

Reviewer 2 Report

Nice work!